# Analysis of the characteristics of the global virtual water trade network using degree and eigenvector centrality, with a focus on food and feed crops

Sang-Hyun Lee[1], Rabi H. Mohtar[1], Jin-Yong Choi[2], Seung-Hwan Yoo[3]

[1]Department of Biological and Agricultural Engineering, Texas A&M University, College station, TX77840, USA
[2]Department of Rural Systems Engineering and Research Institute for Agriculture & Life Sciences, Seoul National University, Seoul, Republic of Korea
[3]Department of Rural and Bio-systems Engineering, Chonnam National University, Gwangju, Republic of Korea

*Correspondence to*: Rabi H. Mohtar (mohtar@tamu.edu)

**Abstract**

This study aims to analyse the characteristics of global virtual water trade (GVWT), such as the connectivity of each trader, vulnerable importers, and influential countries, using degree and eigenvector centrality during the period 2006-2010. The degree centrality was used to measure the connectivity, and eigenvector centrality was used to measure the influence on the entire GVWT network. Mexico, Egypt, China, the Republic of Korea, and Japan were classified as vulnerable importers, because they imported large quantities of virtual water with low connectivity. In particular, Egypt had 15.3 Gm³/year blue water savings effect through GVWT: the vulnerable structure could cause a water shortage problem for the importer. The entire GVWT network could be changed by a few countries, termed "influential traders." We used eigenvector centrality to identify those influential traders. In GVWT for food crops, the U.S.A., Russian Federation, Thailand, and Canada had high eigenvector centrality with large volumes of green water trade. In the case of blue water trade, western Asia, Pakistan, and India had high eigenvector centrality. For feed crops, the green water trade in the U.S.A., Brazil, and Argentina was the most influential. However, Argentina and Pakistan used high proportions of internal water resources for virtual water export (32.9 % and 25.1 %), thus other traders should carefully consider water resource management in these exporters.

Keyword: Virtual water trade; Water footprint; Degree centrality; Eigenvector centrality

## 1 Introduction

Water scarcity is a local phenomenon that is sensitive to global food production, since agriculture has the largest share of the consumption of global freshwater resources (Molden, 2007; Biewald et al., 2014). Most water demand is derived from

agriculture, and crop trade could be considered as the main consumer of water because crop production accompanies water consumption, which is embedded water in crops (Aldaya et al., 2010).

"Virtual water" indicates the embedded water in production and processing (Allan, 1993; Hoekstra, 2003; Yang and Zehnder, 2007), and the virtual water concept has been expanded to include the product chain, and the "water footprint." Building on the virtual water concept, we can convert the crop trade to embedded water trade, called virtual water trade (VWT) (Aldaya et al., 2010). In addition, food security is a significant issue in water-poor regions, because fresh water is a vital factor for growing crops (Konar et al., 2012; Hanjra and Qureshi, 2010; Hoekstra, 2003). The virtual water traded through various crops is regarded as an important variable for global water savings and regional water management, particularly in those regions where water resources are insufficient, such as the Middle East (Hoekstra, 2003; Hoekstra et al., 2011). Accordingly, the concept of VWT brings a new perspective for considering food security, water scarcity, and water resource management together (Novo et al., 2009). In addition, the global virtual water trade (GVWT) could lead to a global redistribution of fresh water and water savings (Konar et al., 2013).

Several studies have been conducted regarding the virtual water trade at different spatial scales, in order to evaluate VWT impacts on water savings (Chapagain et al., 2006). Early studies focused on the water footprint and VWT. Hoekstra and Hung (2005) found that 13% of the total water used for global crop production from 1995 to 1999 was traded internationally, making the international crop trade the main water consumer in importing countries, and causing several researchers to try to estimate virtual water trade. For example, Hanasaki et al. (2010) estimated the global virtual water trade for major crops and livestock products, and Van Oel et al. (2009) quantified the virtual water trade in the Netherlands and evaluated the impact of VWT on water dependency in terms of external water footprint. Bulsink et al. (2010) explained that VWT could increase the resilience to water scarcity in Java, Indonesia. Fader et al. (2011) estimated the internal and external water footprint by VWT and evaluated the effect of VWT on national and global water savings. Therefore, virtual water trade could be the main issue for water security in importing countries such as the Middle East region, and the vulnerable structure of virtual water trade could cause water and food scarcity.

Virtual water trade also has a water scarcity aspect in exporting countries, in terms of water "losses" by exporting commodity (Chapagain et al., 2006). Mubako et al. (2013) calculated water use intensities across economic sectors in California and Illinois, and quantified the water embodied in trade between several states (California, Illinois, and other U.S. states) and the rest of the world. In addition, externalities such as climate change or population change could affect virtual water trade, because virtual water is related to both crop production and water consumption, and the main issue in water resource management is climate change. For example, Konar et al. (2013) quantified the impacts of climate change on virtual water flow and found that the decrease in the total volume of virtual water trade is derived from climate change because of decreased crop trade and virtual water content.

Recently, several studies were conducted to analyse the temporal change of VWT structure using a network system. For example, Konar et al. (2012) analysed the temporal dynamics of virtual water trade networks and found that global food

trade affects water savings and a specific crop network could be more efficient from a water resource perspective. Dalin et al. (2012) focused on the evolution of the GVWT network, considering the number of partners and the volume of virtual water. Generally, studies related to virtual water trade considered more structural change in the entire trade network and the volume of trade in each country. However, we need to understand which countries are vulnerable or influential in GVWT, in order to

5 set a sustainable food trade and water management plan. In addition, crops could be divided into food and feed crops, even if there is not an exact standard for classifying them, because the trade structure of food crops, such as wheat, barley, and rice, have different characteristics from feed crops, such as maize (corn) and beans. The main areas of production and consumption vary greatly according to whether they are food or feed crops. In addition, feed crops are hardly substituted by food crops, and their respective impacts on food security or water security might differ.

This study aims to analyse the characteristics of global virtual water trade (GVWT) of food and feed crops, respectively, through the application of network centrality. Specific objectives are to:

    1.    Evaluate trade vulnerability for each importing country through the connectivity and volume of GVWT.

    2.    Analyse the influential traders of GVWT who could strongly affect the entire trade network.

The degree centrality of the GVWT network was computed to evaluate the connectivity of each country, and a vulnerable

structure in importers indicated low connectivity with a large amount of virtual water imported, potentially causing water shortage problems for importers. We also calculated the eigenvector centrality for measuring the importance and influence of a trader on the whole network, and traders should give pay attention to changes of trade policy and water management of the influential traders.

## 2 Materials and Methods

**2.1 Water footprint (WFP) and global virtual water trade (GVWT)**

Water footprint (WFP, m³/ton) is the volume of water required to produce one ton of crops in the region, and it consists of green and blue water (Hoekstra and Chapagain, 2008). The green water footprint indicates the volume of rainwater consumed, while the blue water footprint indicates the volume of irrigation water (surface and groundwater) consumed. The WFP of a crop indicates the crop water requirement (m³/ha) per yield (kg/ha). It was estimated using Eq. (1), as follows:

$$WFP[c] = \frac{CWR[c]}{Production\ [c]} \qquad (1)$$

where WFP (m³/ton) is the water required for the production of one ton of a given crop c, CWR is the crop water requirement, and the production is the yield per year.

As the water footprint concept, VWT represents the amount of water embedded in products that are traded internationally. Therefore, it was calculated by multiplying the international crop trade by their associated water footprint, and quantifying the global scale of VWT through the water footprint and crops trade using Eq. (2), as follows:

$$VWT[n_e, n_i, c, t] = CT[[n_e, n_i, c, t] \times WFP[n_e, c] \qquad (2)$$

where VWT indicates that VWT from the exporting country $n_e$ to the importing country, $n_i$, CT represents the crop trade, and WFP represents the water footprint. In addition, c and t indicate crop and year, respectively.

## 2.2 Degree centrality of GVWT by network analysis Subsection

GVWT consists of numerous links among nations, and the network approach could be an appropriate method to analyze the structural features of GVWT. In particular, the centrality concept was used to evaluate the main flows and the vulnerable countries. The degree of centrality is one of the simplest indices for evaluating network structure and is a count of the number of edges incident upon a given node (Freeman, 1979). A high level of degree centrality indicates the node has expended connections with various other nodes. The degree centrality has direction and thus, is divided to in-degree and out-degree centrality. In-degree centrality means import in the GVWT network, while out-degree means the opposite. For example, a high level of in-degree centrality in GVWT indicates the country imports virtual water from various exporters, while a high level of out-degree centrality indicates the country exports virtual water to various importers. In other words, a country that has a high level of degree centrality could be identified as a main country in the expanded GVWT network. Therefore, degree centrality could be applied to quantify the connectivity of each country in GVWT. The degree centrality of each country in GVWT is calculated as:

$$C_i = \sum_{j}^{N} VWT_{ij}/(N-1) \tag{3}$$

where $C_i$ is the degree centrality of country i, and N is the number of total countries. $VWT_{ij}$ indicates the virtual water trade between the $i^{th}$ and $j^{th}$ country.

## 2.3 Eigenvector centrality of GVWT by network analysis

GVWT comprises a complex network, but some countries could affect the entire network system: it is important to determine these countries. Thus, we applied eigenvector centrality to the GVWT network in order to find the most influential countries. Eigenvector centrality is used to measure the importance and influence of a node on the whole network (Ruhnau, 2000). The eigenvector centrality represents relative centrality to all nodes in the network based on the principle that high-level centrality nodes could contribute more to connected nodes than low-level centrality nodes. In other words, the centrality of a country not only depends on the number of trade partners adjacent to it, but also on their centrality values (Ruhnau, 2000). Accordingly, the eigenvector centrality could be used to determine influential countries and influence areas. Bonacich (1972) defined the centrality $c(v_i)$ of a node $v_i$ as the positive multiple of the sum of adjacent centralities, as follows:

$$\lambda c(v_i) = \sum_{j=1}^{n} \alpha_{ij} c(v_j) \qquad \forall i. \tag{4}$$

In matrix notation, with $c = (c(v_i), \ldots, c(v_n))$, the above equation yields

$$Ac = \lambda c \tag{5}$$

This type of equation is solved using eigenvalues and eigenvectors. An eigenvector of the maximal eigenvalue with only non-negative entries exists. We call a non-negative eigenvector ($c \geq 0$) of the maximal eigenvalue the principal eigenvector, and we $c(v_i)$ is the eigenvector-centrality of node $v_i$ (Ruhnau, 2000). The eigenvector centrality of a node is proportional to the sum of eigenvector centralities of the connected nodes (Bonacich, 1972). In addition, eigenvector centrality indicates the principal eigenvector that has the largest eigenvalue among all eigenvectors. We used NetMiner 3.0 (http://www.netminer.com) to estimate the degree and the eigenvector centrality.

## 2.4 Data for international trade and water footprint of study crops

In this study, we compared the GVWT of food and feed crops, because food crops, such as wheat and rice, might have different trade characteristics from feed crops, such as maize and soybeans. For example, Konar et al. (2011) found the number of links and average degree of corn and soy were smaller than those of other food crops, such as wheat, barley, and rice.

Although there is no exact classification for food and feed crops, food crops generally indicate crops for food, and representative crops are wheat, barley, and rice. Feed crops indicate crops that are cultivated primarily for animal feed, and the representative crops are maize (corn) and soybeans. In particular, East Asian countries such as China, Japan, and Korea have used maize and beans for animal feed. In this study, food crops included wheat, rice, barley, potatoes, sweet potatoes, rye, and grain sorghum. The feed crops included maize and beans crops. Table 1 lists specific crops.

Country-scale import and export data of various commodities for every 5 years could be obtained from the Personal Computer Trade Analysis System (PC-TAS) produced by the United Nations Statistics Division (UNSD). These data are based on the Commodity Trade Statistics Data Base (COMTRADE) of the UNSD. According to the World Meteorological Organization report (WMO, 2013), there were several significant events related to food trade during 2000-2010. For example, Australia suffered severe drought damage in 2007, but the drought was solved in 2009, and Australia was noticeable as a main exporter in 2010. In addition, the Russian federation had the worst drought, and the government decided to stop exporting wheat, barley, and maize. This action could affect Middle East countries, and also the entire crop trade. We expected the global virtual water trade in these seasons could be important issues, and collected international trade data of food and feed crops during 2006-2010 from PC-TAS.

The water footprint is defined as the total volume of water consumed within the territory of the nation. Mekonnen and Hoekstra (2010) quantified the average values of green and blue water footprints of crops and crop products at national and sub-national levels from 1996 to 2005. The water footprint data indicated the representative index using average value. Therefore, we applied the average value of water footprint during the period 1996-2005 from Mekonnen and Hoekstra (2010), even though this study focused on crop trade from 2006 to 2010.

# 3 Results and Discussion

## 3.1 Estimation of the GVWT of food and feed crops

The GVWT is dependent on the water footprint of each country, and a few countries cultivate and export water intensive crops. The different variability between green and blue water export was derived by the variance of water footprint, which is dependent on the climate features in the exporting country. Mekonnen and Hoekstra (2010) also mentioned the difference of water footprint for each country; for example, relatively smaller water footprints of cereal crops were estimated in Northern and Western Europe than in most parts of Africa. In this study, we showed the variability of green and blue water export, respectively, in crop export during the period 2006-2010 (Fig. 1). The dispersion of scattered points of green water export and crop export was smaller than those of blue water export. One of the reasons why a large dispersion was shown in blue water export might be that the volume of blue water is much smaller than that of green water. Thus, a small amount of blue water might derive a large change in this plot. However, the main issue in Fig. 1 was that the blue water footprint differed more depending on the exporting country, rather than on the green water footprint. Therefore, the variability of blue water export was larger than that of green water export, and crop export could bring differing impacts on irrigation water by country.

In addition, we calculated the total amount of green and blue water trade of each country from 2006 to 2010. For food crops such as wheat, rice, barley etc., the total crop trade between 2006 and 2010 was 985.6 Mton, and the GVWT was 1631.0 Gm³ (green water: 1453.1 Gm³, blue water: 177.9 Gm³). The GVWT of wheat had the highest proportion, totalling 1057.8 Gm³, but the largest amount of blue water was traded by rice. About 136.7 Gm³ of blue water was traded through the rice trade, 4 times higher than that traded through wheat. Barley presented as a less water intensive crop than either wheat or rice. Feed crops between 2006 and 2010, such as maize and beans crops, totaled 1243.8 Mton, with the GVWT at 1811.9 Gm³. The beans crops were representative water intensive crops, and about 1360.4 Gm³ of virtual water was traded between 2006 and 2010. In contrast, the amount of maize traded was 531.2 Mton, but the virtual water that was involved was only 451.5 Gm³.

## 3.2 Analysis of the connectivity and intensity of GVWT using degree centrality

### 3.2.1 Analysis of connectivity in GVWT

The GVWT network includes both the volume of virtual water and the connection among countries. Fig. 2 shows only the main GVWT network of food and feed crops in 2010 using the threshold value of virtual water trade, as we could not display these networks with all links, because it is impossible to distinguish each connection between countries. Therefore, we showed the main links that were over a threshold value of 1.0 Gm³ of total virtual water trade in 2010. Some countries were eliminated from the figure, because they only had connections of virtual water trade that were less than the threshold value. GVWT for food crops has a dispersed network, but GVWT for feed crops is more centralized with a few main exporters,

such as the U.S.A., Argentina, Brazil, and China. In other words, the food and feed crop trades have a different structure, and we need to consider not only volume, but also the connectivity of the virtual water trade.

In this study, degree centrality was applied to understand the connectivity of GVWT. The degree centrality was divided into in- and out- degree by the direction of GVWT. In-degree means imports, and out-degree means exports. We analysed the in-
5 and out-degree centrality of the GVWT of food and feed crops during the period 2006-2010, and Fig. 3 shows the results. The exporters in GVWT for food crops had more connectivity with expanded structure than the exporters in GVWT for feed crops. In addition, importers in the GVWT of the food trade had various connections with exporters.

Considering the out-degree centrality of GVWT for food crops, the U.S.A. displays expanded connectivity with various importers, followed by Asian countries, such as Thailand, Pakistan, Vietnam, and India. Ukraine also had high connectivity
to various importers characterized by large amounts of virtual water export. These countries play the main role for virtual water supply in the GVWT. In contrast, the Russian Federation, Kazakhstan, and Australia had lower connectivity, even though they exported a lot of virtual water by the food crops trade. Considering the out-degree centrality of the GVWT for feed crops, the exporters who exported a lot of virtual water had high connectivity as well. For example, the U.S.A., Brazil, and Argentina had high ranks in both the volume and connectivity of GVWT. These countries exported the largest amount
of virtual water to eastern Asian countries, such as China, Japan, and The Republic of Korea, but also had various connections with importers. Konar et al. (2011) aggregated the virtual water trade of 5 crops and 3 animal products, and measured the node degree of the virtual water trade, which indicated the number of trade partners. They found that the U.S.A., the Netherlands, France, Italy, and the U.K. were the top 5 exporters who had large connections. On the other hand, China and Thailand were the only Asian countries in the top 15 exporters according to the number of connections. However,
in this study, we found that Pakistan, India, and Vietnam also had high connectivity in virtual water export through food crops, because we analysed the connectivity of the virtual water trade of food and feed crops, respectively.

In-degree centrality indicated the connection of virtual water import according to the importer's perspective. Therefore, the importer with a high rank of in-degree centrality imports virtual water from various exporters, meaning that this importer has a robust trade structure. If the importer has a low rank of in-degree centrality with a larger volume of virtual water import,
then this importer might be highly dependent on just a few exporters. For example, Egypt and Japan imported a lot of virtual water by food crops trade, but the rank of in-degree centrality was 21st and 33rd, respectively. Egypt imported over 50% of wheat from only the U.S.A. and Russian Federation. In terms of feed crops trade, most virtual water was imported to China, but the connectivity was very low. In contrast, the Netherlands, Spain, and Germany had high ranks in both the volume and connectivity of virtual water import through the feed crops trade: results indicating that these countries have robust trade
structures. In fact, the European countries have a robust internal trade network with various connections among European countries. Konar et al. (2011) also found that the U.S.A., U.K., Germany, Canada, and Netherlands were the top 5 importers. On the other hand, Saudi Arabia and Hong Kong were the only Asian countries in the top 15 importers. These results are similar in this study; for example, European countries had higher connectivity than Asian countries.

### 3.2.2 Evaluation of vulnerability of virtual water importers through the connectivity and volume of GVWT

The importers in the GVWT were passive by water shortage in exporters, and the GVWT network of importers could be a vulnerable structure by the number of the connections with exporter. For example, in the trend of the increase of crop trade, when the GVWT is concentrated in a few countries, the main exporters could dominate a few importers. This means that

these importers might be dependent on a few exporters with a low resilience structure. Fig. 4 shows the average virtual water import from one exporter. In terms of GVWT for food crops, Mexico imported an average 8.1 Gm³ from one exporter, meaning that Mexico is highly dependent on a few exporters. In the case of feed crops trade, China has the largest average quantity of virtual water imported from one exporter, followed by Mexico and Uruguay. Konar et al. (2011) analysed the strength of each link in the VWT, and found that the link between the U.S.A. and Mexico was the second largest link. In

these importers, virtual water import could be a main issue for sustainable water management, but the VWT, which is highly dependent on a few exporters, could be regarded as a vulnerable trade structure. Therefore, it is important to understand the vulnerability of VWT with consideration of the connectivity and volume of virtual water import.

In this study, the importers of VWT were classified in terms of both connectivity and volume of virtual water import. Tables 2 and 3 classify importing countries according to the volume (I-III) and connectivity (A-C) of GVWT, respectively.

We considered the vulnerability of virtual water trade to be more related to importers with the larger volume of virtual water import. Therefore, the top 10 percentile of total virtual water import was used as the threshold. After that, we classified these countries into 3 groups, according to the top 1, 5, and 10 percentiles of total virtual water import.

The small volume group (I) includes countries that imported above the top 10 percentile and below the top 5 percentile of total virtual water import; the medium group (II) includes the countries that imported over the top 5 percentile and below the

20 top 1 percentile of total virtual water import. Finally, the large volume group (III) includes countries that imported over the top 1 percentile of total virtual water import.

In addition, the vulnerable virtual water trade could be related to the connectivity; therefore, we classified the importers into high, medium, and low connectivity groups, using the degree centrality of links. The importers who have a low degree centrality of links could be regarded as relatively vulnerable importers, and we use the maximum degree centrality of links as

the standard for evaluating the connectivity of each country. Therefore, the importers who have the upper one third of maximum degree centrality are classified as the high connectivity group (C), and the importers who have the lower one third of maximum degree centrality are classified as the low connectivity group (A). The importers who are classified in the medium connectivity group have a degree centrality between the upper one third and lower one third of maximum degree centrality. When importers are classified into the A-III sector, we considered that they had intensive virtual water import

with vulnerable structure.

In food crops, the upper 10% of virtual water import, 25.1 Gm³ was set as the threshold value, and the upper 5 and 1 percentiles of virtual water import were 37.3 and 72.9 Gm³, respectively. Therefore, the importers in the large volume group imported over 72.9 Gm³ of virtual water during 2006-2010 through food crops. The maximum value of degree centrality was

0.22. Therefore, the high connectivity group included those importers who had a degree centrality between 0.22 and 0.15. The low connectivity group included those importers who had lower than 0.07 degree centrality. Mexico, which was located in the A-II sector, was a vulnerable importer in GVWT. In addition, the phenomenon of low in-degree centrality with links of GVWT was shown in Asia countries, even if they imported a lot of virtual water. For example, Iran and the Philippines

were classified into B-II sector, and even Japan was classified into B-III. However, European countries, such as Spain, Turkey, and the Netherlands, were classified into C-I sector. These results represented that the Asian countries imported a lot of virtual water from just a few exporters, and the European countries were connected to various exporters, even if they imported a comparable quantity of virtual water.

In feed crops, the upper 10% of virtual water import, 23.8 Gm³ was set as the threshold value, and the upper 5 and 1

percentiles of virtual water import were 42.3 and 103.6 Gm³, respectively. Therefore, the importers in the large volume group imported over 103.6 Gm³ of virtual water during 2006-2010 through food crops. The maximum value of degree centrality was 0.17. Therefore, the high connectivity group included importers who had a degree centrality between 0.17 and 0.11. The low connectivity group included the importers who had lower than 0.06 degree centrality. Mexico, The Republic of Korea and Germany were in sector II, but Mexico (A-II sector) and The Republic of Korea (B-II sector) had lower

connectivity than that of Germany (C-II sector); that is, Mexico and The Republic of Korea imported large amounts of virtual water from a few countries, and had a vulnerable GVWT structure. In addition, China was regarded as an exclusive importer in the GVWT network. In contrast, European countries, such as the Netherlands, Spain, France, Italy and Germany, had a more distributed structure than eastern Asian countries, who imported large volumes of virtual water by feed crops trade.

**3.2.3 GVWT impacts on water savings in importing countries**

Virtual water trade could help the importers save water resources by crops import. For example, if the importing country replaces crop import with domestic production, this will be accompanied by additional water use. Table 4 shows the water savings by virtual water import in main importers from 2006 to 2010. China and Japan, respectively saved 24.7 and 18.7 Gm³/yr of green water by crops import, while Egypt and Iran, respectively saved 15.3 and 10.1 Gm³/yr of blue water by

crops import, depending on irrigation water for domestic crop production. In particular, Egypt and Iran have few water resources, therefore, the virtual water impacts on water resource savings in these countries might be larger than on other importers.

Accordingly, VWT is a very important issue for these importers; thus the vulnerable structure of VWT could cause water shortage problems to importing countries. For example, in 2010, Russia banned wheat export because of severe drought, and

the global wheat price rose. Oxfam Research Reports analysed the impacts of the Russian ban of wheat export on global and local areas in terms of economic impacts (Welton, 2011). Wheat import in Egypt has high dependency on the Russian federation's export, which we regarded as a vulnerable structure, and the insufficient import of crops due to the export ban in the Russian Federation could bring not only economic impacts but also serious water consumption for increasing domestic

food production. Chapagatin et al. (2006) found the import of wheat in Egypt contributed to a national water saving of 3.6 Gm³/yr during 1997-2001, which according to the 1959 agreement was about seven percent of the total volume of water to which Egypt was entitled. Fader et al. (2011) also found that some water-scarce countries, such as China and Mexico, the Netherlands and Japan, would need relatively high amounts of water to produce the goods they otherwise import: meaning that they save high amounts of water by importing goods. Therefore, if they stopped importing and exporting agricultural products, these countries would need to use more water in their agricultural sectors (Fader et al., 2011). In other words, a vulnerable trade structure with low connectivity could be one of the main reasons for water shortage problems.

### 3.3 Analysis of influential countries in GVWT using Eigenvector centrality

GVWT is complicated to understand and it is difficult to estimate the influence of each trader on GVWT. A country that has relationships with main exporters and importers could influence the GVWT, even if the volume of trade is small. Apart from degree centrality, such a country would have distinctive centrality in terms of the influence on the entire GVWT network. Accordingly, we estimated the eigenvector centrality of green and blue water trade in GVWT, and used degree and eigenvector centrality to analyse the influential importers and exporters. The degree centrality shows the connectivity and volume of the VWT, and the eigenvector centrality shows the influence of countries on the entire GVWT network structure. Therefore, the most influential traders have high degree and eigenvector centrality at the same time, and the other traders should pay attention to changes of trade policy and water management of the influential traders.

Tables 5 and 6 indicate the eigenvector centrality in green and blue water trade, and the degree centrality in connection and volume of the GVWT network, respectively. The U.S.A. showed high out-degree centrality and high eigenvector centrality, which indicates the U.S.A. was the most influential exporter in the green water trade through the food crops trade. The green water trade also had secondary influential exporters, such as Canada, the Russian Federation, Thailand, and Australia. In terms of import, Japan, Mexico and Egypt represented the influential importers for green water trade, and the influence importing area of green water trade was distributed between South America, Europe, western Asia, and East Asia.

In contrast, the influential exporters and importers of the blue water trade differed from the green water trade. The influential global blue water exporters by food crops were the U.S.A., Pakistan, India, and Thailand, while global blue water import was dominated by western Asia, including Iran, Saudi Arabia, and the U.A.E.

For feed crops, the green water in the U.S.A., Brazil, and Argentina was exported to eastern Asian countries, such as China, the Republic of Korea, and Japan. In particular, Brazil and Argentina were dependent on green water. However, the U.S.A. overwhelmingly used a lot of blue water to export maize and beans crops. The U.S.A., Mexico, China, and Japan constructed influential lines from the Americas to eastern Asia.

Crop production is accompanied by water consumption; thus the crop trade could also be affected by the water resource status in the exporting country. Table 7 shows the water resources and virtual water use for domestic crop production and export in the influential countries. In terms of water resources and virtual water use, over 30 % of internal water resources were used for exporting crops in Argentina, followed by Pakistan (25.1%), and the Ukraine (19.4 %).  In addition, some

countries used a lot of water to export crops, for example, over 50 % of virtual water used for food and feed crop production was used for export crops in Argentina, Canada, and Paraguay. In addition, Thailand and Paraguay used 39.5% and 54.2% of domestic virtual water use, respectively, for virtual water export, and the dependence on internal water resources was over 10 % in both countries. Therefore, virtual water export of these countries could be strongly affected by internal water

resources, and this could have a negative impact on importers.

## 4. Conclusions

Crop production is accompanied by water consumption; thus the water resource status in the exporting country could also affect the crop trade. The virtual water trade could help importers save national water resources by importing crops. For example, if the importing country replaced imported crops with domestic production, this would be accompanied by

additional water use. National water savings achieved by the virtual water trade are equal to the import volume multiplied by the volume of water required to domestically produce the commodity. However, the virtual water trade could cause water "losses" for the exporting countries (Chapagain et al., 2006). For example, countries whose major industry is agriculture spend their water resources on the food trade. In addition, the available global freshwater is decreasing due to climate change, suggesting that water should be considered a precious natural resource.

Virtual water trade is the main component for water management for both exporters and importers; thus, it is important to understand the characteristics of virtual water trade. In this study, we used degree and eigenvector centrality to analyse the global virtual water trade (GVWT) during the period 2006-2010, and using the structural characteristics, such as the connectivity of each trader, vulnerable importers, and influential countries. This study only considered the recent 5 years trade, and is limited in terms of prediction. In addition, the global crop trade is related to various factors, such as price,

climate, and policy; thus it is very hard to predict the future trade condition. However, the virtual water concept could provide an extended perspective with which to better understand the food, water, and trade relationship. In particular, importers who had a vulnerable GVWT structure were classified according to their connectivity and volume of GVWT. Mexico, Egypt, China, the Republic of Korea, and Japan were classified as vulnerable importers, because they had low connectivity and imported a lot of virtual water. VWT could bring national water savings, but the vulnerable structure of

VWT could cause problems of water security for importers. For example, Egypt had 15.3 Gm³/year blue water savings effects through GVWT, thus its vulnerable structure could cause water shortage problems.

A few countries that we term influential countries could change the entire GVWT network. In addition, if the influential countries have water shortages, it becomes not only a national scale problem, but also a global threat. Therefore, we classified the influential countries in GVWT using eigenvector centrality, which is generally used to measure influence on an

entire network. For the food crops trade, the influential traders were distinguished by green and blue water trades. For example, the U.S.A., Russian Federation, Thailand, and Canada were classified as influential traders in green water trade. However, western Asia, Pakistan, and India were classified as influential traders in blue water trade. The feed crops trade

was much more dominated by green water than by blue water, and the U.S.A., Brazil, and Argentina were classified as the most influential traders. In particular, Argentina and Pakistan used a high proportion of internal water resources for virtual water export (32.9% and 25.1%, respectively); thus the other traders should consider the water resource management in these exporters carefully. This study could provide information for an integrated global water strategy, and arouse the main

importers attention of the risk of serious dependency on foreign water resources.

**Acknowledgments**

The international trade data are available at a Personal Computer Trade Analysis System (PC-TAS), produced by the United Nations Statistics Division (UNSD). The results data for this study are freely available by contacting the corresponding author.

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

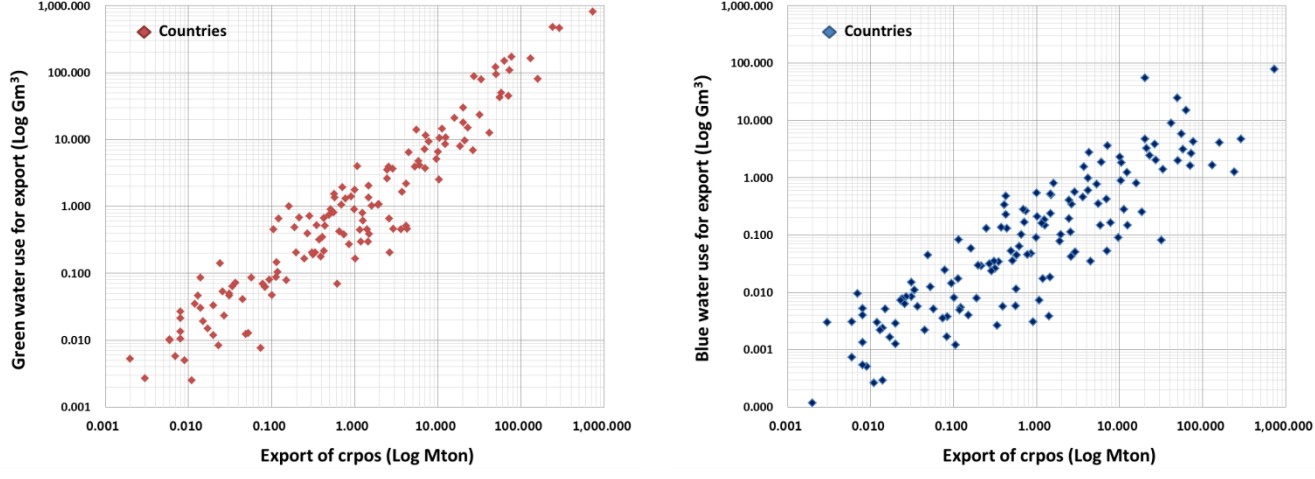

**(a) Crop export and green water export**

**(b) Crop export and blue water export**

**Figure 1: A comparison between virtual water export and crop export during the period 2006-2010 (wheat, barley, rice, rye, sorghum, maize, and beans crops).**

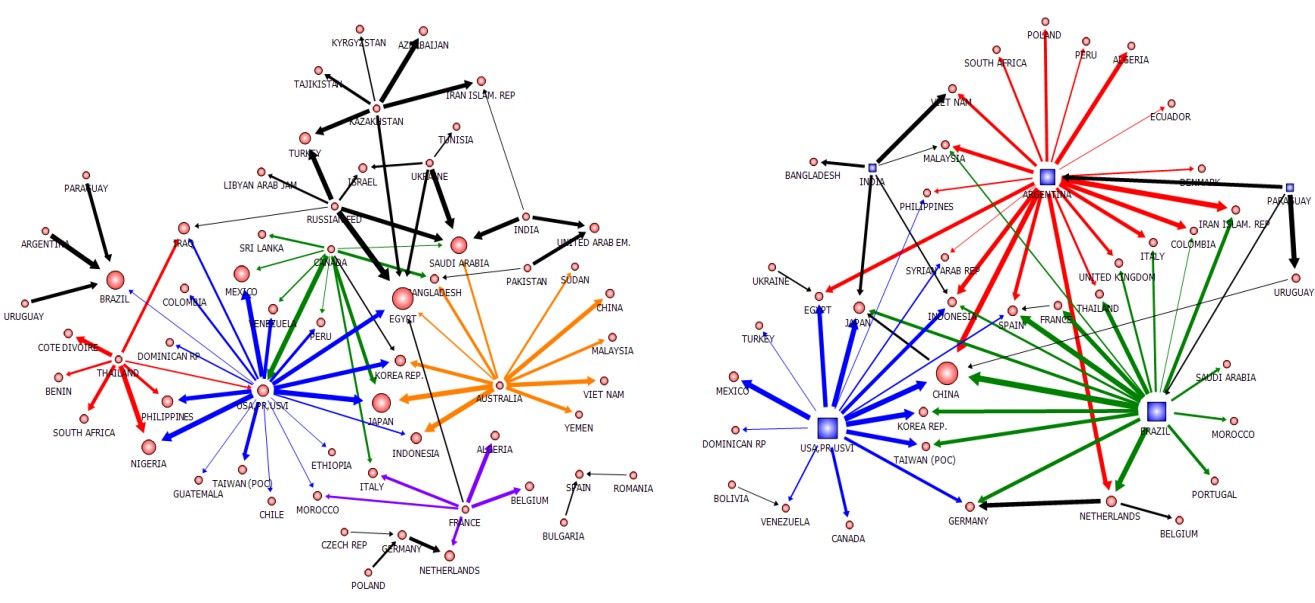

**(a) Food crops (wheat, barley, rice, rye, and sorghum)**

**(b) Feed crops (maize and soybean)**

5   **Figure 2: The GVWT network through food and feed crops trade in 2010.**

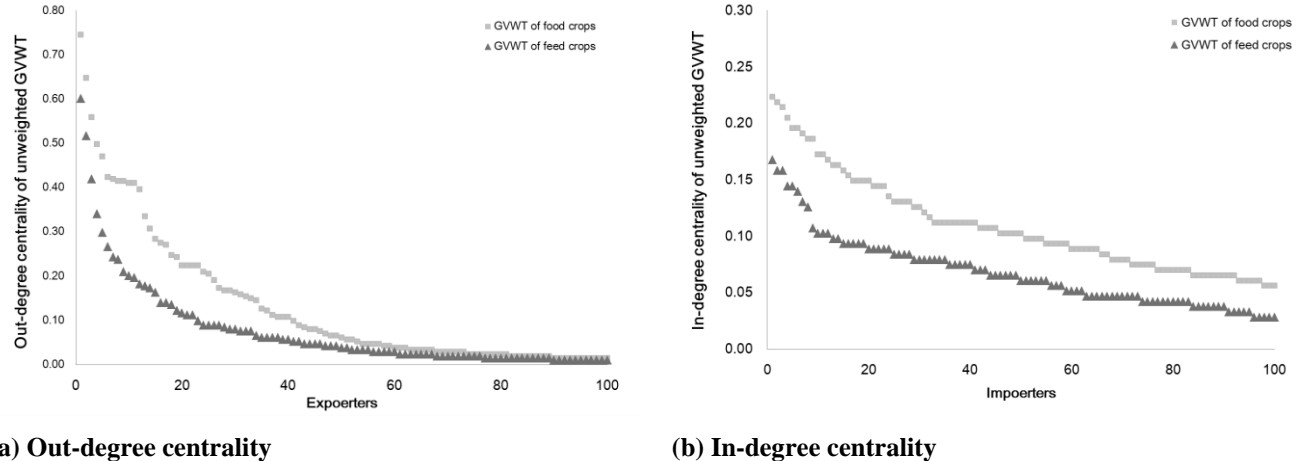

**(a) Out-degree centrality**  **(b) In-degree centrality**

**Figure 3: Out- and in-degree centrality in connection network of GVWT for food and feed crops during the period 2006-2010.**

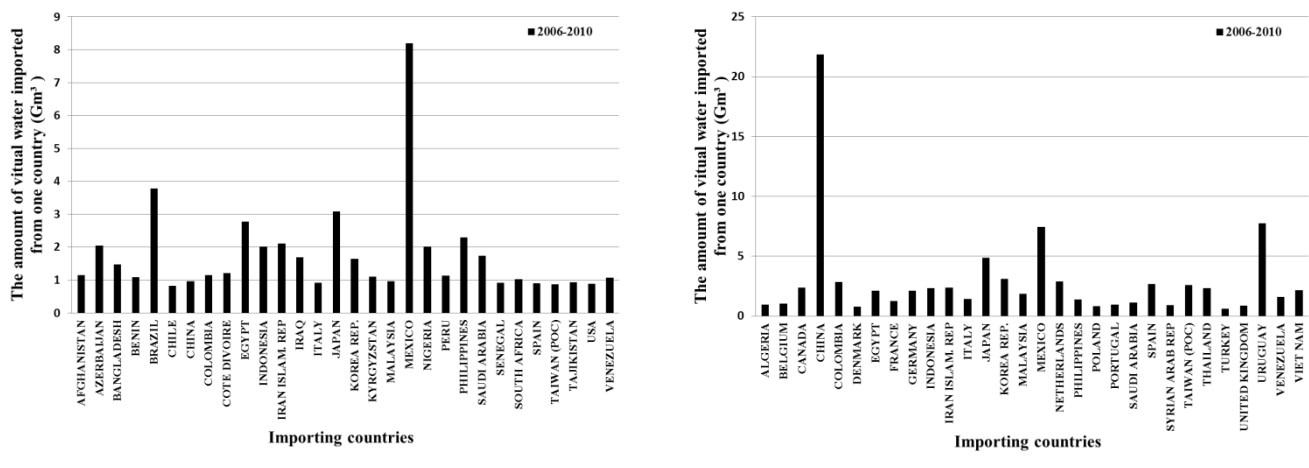

**(a) Food crops (wheat, barley, rice, rye, and sorghum)**  **(b) Feed crops (maize and soybean)**

**Figure 4: The intensive inflow of virtual water by food and feed crops import.**

**Table 1: Study crops for food and feed crops.**

| Crops | | Harmonized System Codes (HS Code) | Description of crop commodity |
|---|---|---|---|
| Food Crops | Wheat | 100190 | Wheat |
| | | 100110 | Durum wheat |
| | Rice | 100610 | Rice in the husk (paddy or rough) |
| | | 100620 | Rice, husked (brown) |
| | | 100630 | Rice, semi-milled or wholly milled |
| | | 100640 | Rice, broken |
| | Barley | 100300 | Barley |
| | Others | 070190 | Potatoes |
| | | 071420 | Sweet potatoes |
| | | 100200 | Rye |
| | | 100700 | Grain sorghum |
| Feed crops | Maize | 100590 | Maize (corn) |
| | | 100510 | Maize (corn) seed |
| | Beans crops | 071332 | Beans, small red (Adzuki) |
| | | 071390 | Leguminous vegetables |
| | | 120100 | Soya beans |
| | | 230400 | Soya-bean oil-cake & solid residues |

**Table 2: Classification of importers by connectivity and volume of GVWT for food crops (wheat, barley, rice, and others).**

| GVWT of food crops | Connectivity of GVWT | | |
|---|---|---|---|
| | Low (A) (lower 1/3 of maximum degree centrality) | Medium (B) (between 1/3 and 2/3 of maximum degree centrality) | High (C) (above 2/3 of maximum degree centrality) |
| **Volume of GVWT** | | | |
| **Small ( I )** (between 10 and 5 percentile) | | BANGLADESH KOREA REP. YEMEN | SPAIN TURKEY USA NETHERLANDS ALGERIA GERMANY UAE |
| **Medium (II)** (between 5 and 1 percentile) | MEXICO | INDONESIA IRAN IRAQ NIGERIA PHILIPPINES BRAZIL | ITALY |
| **Large (III)** (top 1percentile) | | EGYPT JAPAN | SAUDI ARABIA |

**Table 3: Classification of importers by connectivity and volume of GVWT for feed crops (maize and soybean).**

| GVWT of feed crops | | Connectivity of GVWT: Low (A) (lower 1/3 of maximum degree centrality) | Connectivity of GVWT: Medium (B) (between 1/3 and 2/3 of maximum degree centrality) | Connectivity of GVWT: High (C) (above 2/3 of maximum degree centrality) |
|---|---|---|---|---|
| Volume of GVWT | **Small ( Ⅰ )** (between 10 and 5 percentile) | COLOMBIA<br>URUGUAY | TAIWAN<br>IRAN<br>THAILAND<br>VIET NAM<br>EGYPT<br>MALAYSIA | UK |
| | **Medium (Ⅱ)** (between 5 and 1 percentile) | MEXICO | INDONESIA<br>KOREA REP | ITALY<br>FRANCE<br>SPAIN<br>GERMANY |
| | **Large (Ⅲ)** (top 1percentile) | | CHINA<br>JAPAN | NETHERLANDS |

**Table 4: Water resource and virtual water savings by importing crops.**

| **Importers** | **Water resource ($Gm^3$)** | | **VWI* by crop trade ($Gm^3$/yr)** | | **VWU* for producing imported crops ($Gm^3$/yr)** | | **Water savings ($Gm^3$/yr)** | |
|---|---|---|---|---|---|---|---|---|
| | **Internal (1)** | **External (2)** | **Green water (3)** | **Blue water (4)** | **Green water (5)** | **Blue water (6)** | **Green water (3)-(5)** | **Blue water (4)-(6)** |
| CHINA | 221 | 65 | 105.3 | 10.6 | 80.6 | 2.1 | 24.7 | 8.5 |
| EGYPT | 65 | 5 | 3.2 | 16.2 | 23.4 | 0.9 | -20.2 | 15.3 |
| IRAN | 2 | 56 | 9.8 | 11.6 | 15.8 | 1.5 | -6 | 10.1 |
| JAPAN | 409 | 48 | 53.1 | 1.3 | 34.4 | 2.7 | 18.7 | -1.4 |
| MEXICO | 129 | 9 | 36.1 | 5.5 | 21.1 | 2.2 | 15 | 3.3 |

* VWI: virtual water import
* VWU: virtual water use

**Table 5: Eigenvector centrality of green water trade and degree centrality of GVWT.**

| Countries | Eigenvector centrality | In-degree centrality | | Out-degree centrality | |
|---|---|---|---|---|---|
| | Green water trade | Volume of GVWT | Connection of GVWT | Volume of GVWT | Connection of GVWT |
| **GVWT for food crops** | | | | | |
| USA | 0.62 | 0.14 | 0.16 | 1.64 | 0.74 |
| Japan | 0.34 | 0.34 | 0.11 | 0.00 | 0.11 |
| Canada | 0.29 | 0.02 | 0.08 | 0.68 | 0.41 |
| Mexico | 0.28 | 0.23 | 0.03 | 0.02 | 0.08 |
| Egypt | 0.23 | 0.40 | 0.14 | 0.01 | 0.24 |
| Nigeria | 0.23 | 0.23 | 0.12 | 0.00 | 0.01 |
| Russian Federation | 0.17 | 0.05 | 0.13 | 0.81 | 0.41 |
| Thailand | 0.17 | 0.05 | 0.11 | 0.66 | 0.65 |
| Philippines | 0.15 | 0.21 | 0.09 | 0.00 | 0.00 |
| Iraq | 0.13 | 0.17 | 0.10 | 0.00 | 0.00 |
| Korea Rep. | 0.12 | 0.15 | 0.09 | 0.00 | 0.01 |
| Indonesia | 0.11 | 0.19 | 0.09 | 0.00 | 0.03 |
| Australia | 0.10 | 0.01 | 0.09 | 0.44 | 0.28 |
| **GVWT for feed crops** | | | | | |
| China | 0.62 | 1.83 | 0.08 | 0.10 | 0.17 |
| USA | 0.47 | 0.03 | 0.10 | 2.49 | 0.60 |
| Brazil | 0.45 | 0.07 | 0.02 | 2.16 | 0.42 |
| Argentina | 0.26 | 0.06 | 0.04 | 1.78 | 0.52 |
| Japan | 0.17 | 0.52 | 0.11 | 0.00 | 0.01 |
| Netherlands | 0.15 | 0.48 | 0.17 | 0.20 | 0.20 |
| Mexico | 0.11 | 0.31 | 0.04 | 0.01 | 0.06 |
| Spain | 0.10 | 0.39 | 0.14 | 0.02 | 0.07 |
| Korea Rep. | 0.10 | 0.30 | 0.10 | 0.00 | 0.03 |

**Table 6: Eigenvector and degree centrality of blue water trade and degree centrality of GVWT.**

| Countries | Eigenvector centrality of Blue water trade | In-degree centrality | | Out-degree centrality | |
|---|---|---|---|---|---|
| | | Volume of GVWT | Connection of GVWT | Volume of GVWT | Connection of GVWT |
| **GVWT for food crops** | | | | | |
| Pakistan | 0.63 | 0.05 | 0.10 | 0.33 | 0.56 |
| UAE | 0.38 | 0.12 | 0.17 | 0.01 | 0.12 |
| Iran | 0.27 | 0.22 | 0.10 | 0.01 | 0.05 |
| USA | 0.22 | 0.14 | 0.16 | 1.64 | 0.74 |
| Kenya | 0.19 | 0.06 | 0.15 | 0.00 | 0.04 |
| Afghanistan | 0.17 | 0.04 | 0.03 | 0.00 | 0.00 |
| Saudi Arabia | 0.17 | 0.34 | 0.20 | 0.00 | 0.04 |
| Thailand | 0.16 | 0.05 | 0.11 | 0.66 | 0.65 |
| India | 0.16 | 0.07 | 0.13 | 0.24 | 0.47 |
| Mozambique | 0.13 | 0.04 | 0.09 | 0.00 | 0.02 |
| South Africa | 0.11 | 0.11 | 0.11 | 0.01 | 0.07 |
| Mexico | 0.11 | 0.23 | 0.03 | 0.02 | 0.08 |
| Iraq | 0.10 | 0.17 | 0.10 | 0.00 | 0.00 |
| Philippines | 0.10 | 0.21 | 0.09 | 0.00 | 0.00 |
| Oman | 0.10 | 0.03 | 0.11 | 0.00 | 0.03 |
| **GVWT for feed crops** | | | | | |
| USA | 0.70 | 0.03 | 0.10 | 2.49 | 0.60 |
| China | 0.49 | 1.83 | 0.08 | 0.10 | 0.17 |
| Japan | 0.38 | 0.52 | 0.11 | 0.00 | 0.01 |
| Mexico | 0.26 | 0.31 | 0.04 | 0.01 | 0.06 |
| Korea Rep. | 0.16 | 0.30 | 0.10 | 0.00 | 0.03 |
| Taiwan | 0.12 | 0.19 | 0.07 | 0.00 | 0.04 |

**Table 7: Water resource and virtual water use for production and exporting crops.**

| Exporters | Water resource (Gm³) | | VWU* for crop production (Gm³/yr) | | VWE* by crop trade (Gm³/yr) | | Proportion of VWE* (%) | |
|---|---|---|---|---|---|---|---|---|
| | Internal (1) | External (2) | Green water (3) | Blue water (4) | Green water (5) | Blue water (6) | {(5)+(6)}/(1) | {(5)+(6)}/{(3)+(4)} |
| ARGENTINA | 276 | 538 | 140.6 | 1.2 | 90.5 | 0.4 | 32.9 | 64.1 |
| BRAZIL | 5,418 | 2,815 | 213.5 | 0.1 | 92.8 | 0.0 | 1.7 | 43.5 |
| CANADA | 2,850 | 52 | 42.5 | 0.2 | 28.7 | 0.1 | 1.0 | 67.5 |
| FRANCE | 200 | 11 | 34.2 | 1.6 | 15.4 | 0.6 | 8.0 | 44.9 |
| PAKISTAN | 55 | 192 | 21.2 | 53.2 | 3.2 | 10.6 | 25.1 | 18.6 |
| PARAGUAY | 94 | 242 | 19.1 | 0.0 | 10.4 | 0.0 | 11.0 | 54.2 |
| RUSSIAN FED | 4,313 | 195 | 168.5 | 4.2 | 33.9 | 0.5 | 0.8 | 19.9 |
| THAILAND | 226 | 214 | 59.4 | 12.1 | 23.4 | 4.8 | 12.6 | 39.5 |
| UKRAINE | 53 | 86 | 48.1 | 0.7 | 9.9 | 0.4 | 19.4 | 21.1 |
| USA | 2,818 | 251 | 423.7 | 42.8 | 162.3 | 15.0 | 6.3 | 38.0 |

* VWU: virtual water use
* VWE: virtual water export