# Peer review of "Analysis of the characteristics of the global virtual water trade network using degree and eigenvector centrality, with a focus on food and feed crops"

_Hydrology and Earth System Sciences, 2016_

## Referee Comment (RC1) · Anonymous Referee #1 · 19 Jun 2016

This paper presents an interesting study of using network system analysis with the concept of centrality to evaluate the influential and vulnerable countries in global virtual water trade system. The manuscript is on topic is of interest to the journal, but the writing is confusing at parts.

Specific comments:

1. P3, L24: Years 2006∼2010 are not really recent. Maybe add an explanation about the selection of the study period.

2. P4, L1: Maybe delete "Subsection (as Heading 2)".

[Figure]

3. P4, L13~15: I assume a node represents a country here, so maybe use "country", instead of "node" to explain the equation to avoid confusion. Also, please delete the comma at the end of equation 3.

4. P5, L9~11: I would think the fact that blue water consumption for crop export is smaller than green water consumption is part of the reason that green water export has a stronger correlation with crop export.

5. P6, Figure 1: Typos, change "crpos" to "crops".

6. P6, L5: Typo, change "paly" to "play".

7. P6, L1~20: For this part of the discussion on specific countries' high and low connectivity in the virtual water trade system, the authors may need to add some reference to support their statements.

8. P7, L15~23: In this part, the authors suggested that virtual water imports have saved water resources in several countries and the vulnerable structure of VWT could cause water shortage problems in importing countries. Please add references to support these statements.

9. Table 4 and 5: The numbers in the "GVWT for feed crops" part of the two tables are identical. Could the authors explain the reason for this?

10. P8, L19~22: The words "However" and "but" in this part make the logic hard to follow. Please revise.

---

## Referee Comment (RC2) · Anonymous Referee #2 · 2 Jul 2016

This paper presents an interesting network analysis of global virtual water trade on food crops and feed crops. However it is not clear why it is important to focus on food crops and feed crops, when some other papers (e.g. Konar et al. (2011), Konar et al. (2012)) have already discussed the network characteristics of global virtual water trade on total crops and specific crop types. Also, it is not clear how the results are comparing to previous papers. More discussion is needed.

Major comments:

1. In introduction, the author provides a good review on previously literature. However

it is not clear how this paper builds on this rich literature. In particular, how does this paper contribute to the literature? What's the novelty of the paper? Why it is vital to look at food crops and feed crops?

2. In discussion part, more deep analysis comparing your results with other papers is needed. Are the results similar as those in previous papers? Are there some papers to support your conclusions? Are there some unique features in network of food crops and feed crops, when comparing to that of total crops? Why they are different?

For example, P5 L12-19 discusses GVWTs by crops, which have been calculated in Table 3 in Konar et al. (2011). It might not be helpful to replicate previous works.

Minor comments:

1. Please define "food crops" and "feed crops". What specific crop types do they cover?

2. Section 2: Please add one subsection about data source, and provide more details. For example, what is the unit of the data? What commodities the trade data provide?

3. P5 L9-11: This make no sense. When calculating virtual water trade (VWT), we multiply CT by WFP (i.e. Equation (2) in P3 L20). Take the log, and we get log(VWT) = log(CT) + log(WFP) In Figure 1, the slope between log(VWT) and log(CT) should be 1. The only reason for the diffusion is log(WFP), which is dependent on climate features in exporting country. The diffusion in Figure 1(b) is larger than that in Figure 1(a). That is because the variance of WFP for blue water is larger than that for green water.

4. Fig 2: It seems that a subset of countries and links is plotted. Please clarify how you select those countries and links.

5. P6 L33: Please clarify how the volume and connectivity are classified into three groups. In particular, what is the threshold? Why the threshold is as it is?

6. P7 L15-22: Discussion about water savings is irrelevant to this part, which focuses on vulnerability. Please take it out.

Reference:

Konar, M. , C. Dalin, S. Suweis, N. Hanasaki, A. Rinaldo and I. Rodriguez-Iturbe (2011), Water for food: The global virtual water trade network, Water Resources Research, Vol 47, Issue 5, W05520, doi: 10.1029/2010WR010307.

Konar, M. , C. Dalin, N. Hanasaki, A. Rinaldo and I. Rodriguez-Iturbe (2012), Temporal dynamics of blue and green virtual water trade networks, Water Resources Research, Vol 48, Issue 7, W07509, doi: 10.1029/2012WR011959.

---

## Author Comment (AC1) · 8 Aug 2016

We appreciate the feedback and comments and we believe that these comments improved this paper. We revised the previous paper with reflection of reviews' comments. In addition, we made a revision note for explaining what we revised. Please check the attached file; supplement.zip. including revised manuscript, revision note, and supplement. Thank you again for your comments.

Please also note the supplement to this comment:

[Figure]

http://www.hydrol-earth-syst-sci-discuss.net/hess-2016-211/hess-2016-211-AC1-supplement.zip

---

## Author Comment (AC2) · 8 Aug 2016

We appreciate the feedback and comments and we believe that these comments improved this paper. We revised the previous paper with reflection of reviews' comments. In addition, we made a revision note for explaining what we revised. Please check the attached file; supplement.zip. including revised manuscript, revision note, and supplement. Thank you again for your comments.

Please also note the supplement to this comment:

http://www.hydrol-earth-syst-sci-discuss.net/hess-2016-211/hess-2016-211-AC2-supplement.zip

---

## Author Response (AR1)

**Dear Editor and Reviewers**

We revised the previous paper with reflection of reviewer's comments.
Most of the comments were related to the novelty of this study, comparison to other research, and more explanation for clarifying the results.

1) **Novelty and contribution**
➔ We added more explanation about literature reviews. In addition, we tried to mention the novelty of this paper with a focus on the connectivity, vulnerable importers, and influential countries in the GVWT at Introduction part.

2) **Data collection**
➔ We tried to show the importance of division into food and feed crops in GVWT. We added the sub-section about data source, and explained the types of crops, data source, and unit of the data in this section.
2.4 Data for international trade and water footprint of study crops.

3) **More references for supporting statements about connectivity and vulnerability of GVWT**
➔ We tried to explain deeper discussion with comparison to other research, and added the more explanation.

4) **Other comments**
➔ We tried to reflect reviewer's comments related to clarification of results, study period, and study crops

We tried to incorporate our responses to these comments within the new manuscript.
We appreciate the feedback and comments and we believe that these comments improved this paper.

**Reviewer #1**

| Comments 1 | | |
|---|---|---|
| Reviewer's comments | | P3, L24: Years 2006_2010 are not really recent. Maybe add an explanation about the selection of the study period. |
| Response | | We added the new section about data and explained the selection of the study period. |
| | page 5, line 19 - page 5, line 25 | According to the World Meteorological Organization report (WMO, 2013), there were several significant events related to food trade during 2000-2010. For example, Australia suffered severe drought damage in 2007, but the drought was solved in 2009, and Australia was noticeable as a main exporter in 2010. In addition, the Russian federation had the worst drought, and the government decided to stop exporting wheat, barley, and maize. This action could affect Middle East countries, and also the entire crop trade. We expected the global virtual water trade in these seasons could be important issues, and collected international trade data of food and feed crops during 2006-2010 from PC-TAS. |

| Comments 2 | |
|---|---|
| Reviewer's comments | P4, L1: Maybe delete "Subsection (as Heading 2)". |
| Response | We removed "as Heading 2" |

| Comments 3 | |
|---|---|
| Reviewer's comments | P4, L13~15: I assume a node represents a country here, so maybe use "country", instead of "node" to explain the equation to avoid confusion. Also, please delete the comma at the end of equation. |
| Response | We changed "node" to "country" with following your comments, and deleted the comma. |

| Comments 4 | | |
|---|---|---|
| Reviewer's comments | | P5, L9~11: I would think the fact that blue water consumption for crop export is smaller than green water consumption is part of the reason that green water export has a stronger correlation with crop export. |
| Response | | We tried to explain the diffusion of green and blue water export rather than the difference of amount of virtual water export. The different diffusion between green and blue water export was derived by the variance of water footprint, which is dependent on climate features in exporting country. Therefore, we changed the explanation about Figure 1 and focused on the diffusion of green and blue water export. |
| | page 6, line 3 - page 6, line 14 | The GVWT is dependent on the water footprint of each country, and a few countries cultivate and export water intensive crops. The different variability between green and blue water export was derived by the variance of water footprint, which is dependent on the climate features in the exporting country. Mekonnen and Hoekstra (2010) also mentioned the difference of water footprint for each country; for example, relatively smaller water footprints of cereal crops were estimated in Northern and Western Europe than in most parts of Africa. In this study, we showed the variability of green and blue water export, respectively, in crop export during the period 2006-2010 (Fig. 1). The dispersion of scattered points of green water export and crop export was smaller than those of blue water export. One of the reasons why a large dispersion was shown in blue water export might be that the volume of blue water is much smaller than that of green water. Thus, a small amount of blue water might derive a large change in this plot. However, the main issue in Fig. 1 was that the blue water footprint differed more depending on the exporting country, rather than on the green water footprint. Therefore, the variability of blue water export was larger than that of green water export, and crop export could bring differing impacts on irrigation water by country. |

**Comments 5**

| Reviewer's comments | P6, Figure 1: Typos, change "crpos" to "crops". |
|---|---|
| Response | We corrected typos. |

**Comments 6**

| Reviewer's comments | P6, L5: Typo, change "paly" to "play". |
|---|---|
| Response | We corrected typos. |

**Comments 7**

| Reviewer's comments | | P6, L1～20: For this part of the discussion on specific countries' high and low connectivity in the virtual water trade system, the authors may need to add some reference to support their statements. |
|---|---|---|
| Response | | We considered the results from Konar et al (2011), and compared to the results of this study. We added the more explanation. |
| | page 7, line 16 - page 7, line 21 | Konar et al. (2011) aggregated the virtual water trade of 5 crops and 3 animal products, and measured the node degree of the virtual water trade, which indicated the number of trade partners. They found that the U.S.A., the Netherlands, France, Italy, and the U.K. were the top 5 exporters who had large connections. On the other hand, China and Thailand were the only Asian countries in the top 15 exporters according to the number of connections. However, in this study, we found that Pakistan, India, and Vietnam also had high connectivity in virtual water export through food crops, because we analysed the connectivity of the virtual water trade of food and feed crops, respectively. |
| | page 7, line 31 - page 7, line 33 | Konar et al. (2011) also found that the U.S.A., U.K., Germany, Canada, and Netherlands were the top 5 importers. On the other hand, Saudi Arabia and Hong Kong were the only Asian countries in the top 15 importers. These results are similar in this study; for example, European countries had higher connectivity than Asian countries. |

| Comments 8 | |
|---|---|
| Reviewer's comments | P7, L15～23: In this part, the authors suggested that virtual water imports have saved water resources in several countries and the vulnerable structure of VWT could cause water shortage problems in importing countries. Please add references to support these statements. |
| Response | We defined the vulnerable structure of VWT and considered this structure could cause water shortage problem in this study. For example, in 2010, Russia banned the wheat export because of severe drought, and global wheat price went up. Oxfam Research Reports analyzed the impacts of Russia ban of wheat export on global and local area in terms of economic impacts (Welton, 2011). However, it was hard to find the reports about relationship between water shortage and virtual water trade. Accordingly, we referenced the studies about water saving impacts in importing countries through trade, and tried to explain the vulnerable trade could cause the decrease of the water saving impacts. |

| | page 9, line 20 - page 10, line 7 | **3.2.3 GVWT impacts on water savings in importing countries**
Virtual water trade could help the importers save water resources by crops import. For example, if the importing country replaces crop import with domestic production, this will be accompanied by additional water use. Table 4 shows the water savings by virtual water import in main importers from 2006 to 2010. China and Japan, respectively saved 24.7 and 18.7 Gm³/yr of green water by crops import, while Egypt and Iran, respectively saved 15.3 and 10.1 Gm³/yr of blue water by crops import, depending on irrigation water for domestic crop production. In particular, Egypt and Iran have few water resources, therefore, the virtual water impacts on water resource savings in these countries might be larger than on other importers.
Accordingly, VWT is a very important issue for these importers; thus the vulnerable structure of VWT could cause water shortage problems to importing countries. For example, in 2010, Russia banned wheat export because of severe drought, and the global wheat price rose. Oxfam Research Reports analysed the impacts of the Russian ban of wheat export on global and local areas in terms of economic impacts (Welton, 2011). Wheat import in Egypt has high dependency on the Russian federation's export, which we regarded as a vulnerable structure, and the insufficient import of crops due to the export ban in the Russian Federation could bring not only economic impacts but also serious water consumption for increasing domestic food production. Chapagatin et al. (2006) found the import of wheat in Egypt contributed to a national water saving of 3.6 Gm³/yr during 1997-2001, which according to the 1959 agreement was about seven percent of the total volume of water to which Egypt was entitled. Fader et al. (2011) also found that some water-scarce countries, such as China and Mexico, the Netherlands and Japan, would need relatively high amounts of water to produce the goods they otherwise import: meaning that they save high amounts of water by importing goods. Therefore, if they stopped importing and exporting agricultural products, these countries would need to use more water in their agricultural sectors (Fader et al., 2011). In other words, a vulnerable trade structure with low connectivity could be one of the main reasons for water shortage problems. |
|---|---|---|

| Comments 9 | |
|---|---|
| Reviewer's comments | Table 4 and 5: The numbers in the "GVWT for feed crops" part of the two tables are identical. Could the authors explain the reason for this? |
| Response | This was a mistake when I copied table to manuscript.
I revised the table. |

| Comments 10 | | |
|---|---|---|
| Reviewer's comments | | P8, L19～22: The words "However" and "but" in this part make the logic hard to follow. Please revise. |
| Response | | We revised these sentences. |
| | page 10, line 32 - page 11, line 5 | In terms of water resources and virtual water use, over 30 % of internal water resources were used for exporting crops in Argentina, followed by Pakistan (25.1%), and the Ukraine (19.4 %).  In addition, some countries used a lot of water to export crops, for example, over 50 % of virtual water used for food and feed crop production was used for export crops in Argentina, Canada, and Paraguay. In addition, Thailand and Paraguay used 39.5% and 54.2% of domestic virtual water use, respectively, for virtual water export, and the dependence on internal water resources was over 10 % in both countries. Therefore, virtual water export of these countries could be strongly affected by internal water resources, and this could have a negative impact on importers. |

**Reviewer #2**

| Major Comments 1 | | |
|---|---|---|
| Reviewer's comments | | In introduction, the author provides a good review on previously literature. However it is not clear how this paper builds on this rich literature. In particular, how does this paper contribute to the literature? What's the novelty of the paper? Why it is vital to look at food crops and feed crops? |
| Response | | We added more explanation about literature reviews. In addition, we tried to mention the novelty of this paper with a focus on the connectivity, vulnerable importers, and influential countries in the GVWT. We tried to show the importance of division into food and feed crops in GVWT, and added crops information in new chapter; 2.4 Data for international trade and water footprint of study crops. |
| | page 3, line 3 - page 3, line 18 | Generally, studies related to virtual water trade considered more structural change in the entire trade network and the volume of trade in each country. However, we need to understand which countries are vulnerable or influential in GVWT, in order to set a sustainable food trade and water management plan. In addition, crops could be divided into food and feed crops, even if there is not an exact standard for classifying them, because the trade structure of food crops, such as wheat, barley, and rice, have different characteristics from feed crops, such as maize (corn) and beans. The main areas of production and consumption vary greatly according to whether they are food or feed crops. In addition, feed crops are hardly substituted by food crops, and their respective impacts on food security or water security might differ. This study aims to analyse the characteristics of global virtual water trade (GVWT) of food and feed crops, respectively, through the application of network centrality. Specific objectives are to: 1. Evaluate trade vulnerability for each importing country through the connectivity and volume of GVWT. 2. Analyse the influential traders of GVWT who could strongly affect the entire trade network. The degree centrality of the GVWT network was computed to evaluate the connectivity of each country, and a vulnerable structure in importers indicated low connectivity with a large amount of virtual water imported, potentially causing water shortage problems for importers. We also calculated the eigenvector centrality for measuring the importance and influence of a trader on the whole network, and traders should give pay attention to changes of trade policy and water management of the influential traders. |

| Major Comments 2 | |
|---|---|
| Reviewer's comments | In discussion part, more deep analysis comparing your results with other papers is needed. Are the results similar as those in previous papers? Are there some papers to support your conclusions? Are there some unique features in network of food crops and feed crops, when comparing to that of total crops? Why they are different? For example, P5 L12-19 discusses GVWTs by crops, which have been calculated in Table 3 in Konar et al. (2011). It might not be helpful to replicate previous works. |
| Response | We tried to explain deeper discussion with comparison to other research. |
| page 7, line 8 - page 7, line 33 | Considering the out-degree centrality of GVWT for food crops, the U.S.A. displays expanded connectivity with various importers, followed by Asian countries, such as Thailand, Pakistan, Vietnam, and India. Ukraine also had high connectivity to various importers characterized by large amounts of virtual water export. These countries play the main role for virtual water supply in the GVWT. In contrast, the Russian Federation, Kazakhstan, and Australia had lower connectivity, even though they exported a lot of virtual water by the food crops trade. Considering the out-degree centrality of the GVWT for feed crops, the exporters who exported a lot of virtual water had high connectivity as well. For example, the U.S.A., Brazil, and Argentina had high ranks in both the volume and connectivity of GVWT.  These countries exported the largest amount of virtual water to eastern Asian countries, such as China, Japan, and The Republic of Korea, but also had various connections with importers. Konar et al. (2011) aggregated the virtual water trade of 5 crops and 3 animal products, and measured the node degree of the virtual water trade, which indicated the number of trade partners. They found that the U.S.A., the Netherlands, France, Italy, and the U.K. were the top 5 exporters who had large connections. On the other hand, China and Thailand were the only Asian countries in the top 15 exporters according to the number of connections. However, in this study, we found that Pakistan, India, and Vietnam also had high connectivity in virtual water export through food crops, because we analysed the connectivity of the virtual water trade of food and feed crops, respectively. In-degree centrality indicated the connection of virtual water import according to the importer's perspective. Therefore, the importer with a high rank of in-degree centrality imports virtual water from various exporters, meaning that this importer has a robust trade structure. If the importer has a low rank of in-degree centrality with a larger volume of virtual water import, then this importer might be highly dependent on just a few exporters. For example, Egypt and Japan imported a lot of virtual water by food crops trade, but the rank of in-degree centrality was 21st and 33rd, respectively. Egypt imported over 50% of wheat from only the U.S.A. and Russian Federation. In terms of feed crops trade, most virtual water was imported to China, but the connectivity was very low. In contrast, the Netherlands, Spain, and Germany had high ranks in both the volume and connectivity of virtual water import through the feed crops trade: results indicating that these countries have robust trade structures. In fact, the European countries have a robust internal trade network with various connections among European countries. Konar et al. (2011) also found that the U.S.A., U.K., Germany, Canada, and Netherlands were the top 5 importers. On the other hand, Saudi Arabia and Hong Kong were the only Asian countries in the top 15 importers. These results are similar in this study; for example, European countries had higher connectivity than Asian countries. |

| Minor Comments 1-2 | |
|---|---|
| Reviewer's comments | Please define "food crops" and "feed crops".
 What specific crop types do they cover?
 Section2: Please add one subsection about data source, and provide more details.
 For example, what is the unit of the data? What commodities the trade data provide? |
| Response | We added the sub-section about data source, and explained the types of crops, data source, and unit of the data in this section. |

| | page 5, line 7 - page 5, line 30 | **2.4 Data for international trade and water footprint of study crops**
 In this study, we compared the GVWT of food and feed crops, because food crops, such as wheat and rice, might have different trade characteristics from feed crops, such as maize and soybeans. For example, Konar et al. (2011) found the number of links and average degree of corn and soy were smaller than those of other food crops, such as wheat, barley, and rice. Although there is no exact classification for food and feed crops, food crops generally indicate crops for food, and representative crops are wheat, barley, and rice. Feed crops indicate crops that are cultivated primarily for animal feed, and the representative crops are maize (corn) and soybeans. In particular, East Asian countries such as China, Japan, and Korea have used maize and beans for animal feed. In this study, food crops included wheat, rice, barley, potatoes, sweet potatoes, rye, and grain sorghum. The feed crops included maize and beans crops. Table 1 lists specific crops.
 Country-scale import and export data of various commodities for every 5 years could be obtained from the Personal Computer Trade Analysis System (PC-TAS) produced by the United Nations Statistics Division (UNSD). These data are based on the Commodity Trade Statistics Data Base (COMTRADE) of the UNSD. According to the World Meteorological Organization report (WMO, 2013), there were several significant events related to food trade during 2000-2010. For example, Australia suffered severe drought damage in 2007, but the drought was solved in 2009, and Australia was noticeable as a main exporter in 2010. In addition, the Russian federation had the worst drought, and the government decided to stop exporting wheat, barley, and maize. This action could affect Middle East countries, and also the entire crop trade. We expected the global virtual water trade in these seasons could be important issues, and collected international trade data of food and feed crops during 2006-2010 from PC-TAS.
 The water footprint is defined as the total volume of water consumed within the territory of the nation. Mekonnen and Hoekstra (2010) quantified the average values of green and blue water footprints of crops and crop products at national and sub-national levels from 1996 to 2005. The water footprint data indicated the representative index using average value. Therefore, we applied the average value of water footprint during the period 1996-2005 from Mekonnen and Hoekstra (2010), even though this study focused on crop trade from 2006 to 2010. |
|---|---|---|

**Table 1: Study crops for food and feed crops.**

| Crops | | Harmonized System Codes (HS Code) | Description of crop commodity |
|---|---|---|---|
| Food Crops | Wheat | 100190 | Wheat |
| | | 100110 | Durum wheat |
| | Rice | 100610 | Rice in the husk (paddy or rough) |
| | | 100620 | Rice, husked (brown) |
| | | 100630 | Rice, semi-milled or wholly milled |
| | | 100640 | Rice, broken |
| | Barley | 100300 | Barley |
| | Others | 070190 | Potatoes |
| | | 071420 | Sweet potatoes |
| | | 100200 | Rye |
| | | 100700 | Grain sorghum |
| Feed crops | Maize | 100590 | Maize (corn) |
| | | 100510 | Maize (corn) seed |
| | Beans crops | 071332 | Beans, small red (Adzuki) |
| | | 071390 | Leguminous vegetables |
| | | 120100 | Soya beans |
| | | 230400 | Soya-bean oil-cake & solid residues |

| Minor Comments 3 | |
|---|---|
| Reviewer's comments | P5 L9-11: This makes no sense. When calculating virtual water trade (VWT), we multiply CT by WFP (i.e. Equation (2) in P3 L20). Take the log, and we get log(VWT) = log(CT) + log(WFP) In Figure 1, the slope between log(VWT) and log(CT) should be 1.
The only reason for the diffusion is log(WFP), which is dependent on climate features in exporting country.
The diffusion in Figure 1(b) is larger than that in Figure 1(a). That is because the variance of WFP for blue water is larger than that for green water. |
| Response | We tried to explain the variability of green and blue water export rather than the volume of virtual water export. The different variability between green and blue water export was derived by the variance of water footprint, which is dependent on climate features in exporting country.
Therefore, we changed the explanation about Figure 1 and focused on the variability of green and blue water export. |
| page 6, line 3 - page 6, line 14 | The GVWT is dependent on the water footprint of each country, and a few countries cultivate and export water intensive crops. The different variability between green and blue water export was derived by the variance of water footprint, which is dependent on the climate features in the exporting country. Mekonnen and Hoekstra (2010) also mentioned the difference of water footprint for each country; for example, relatively smaller water footprints of cereal crops were estimated in Northern and Western Europe than in most parts of Africa. In this study, we showed the variability of green and blue water export, respectively, in crop export during the period 2006-2010 (Fig. 1). The dispersion of scattered points of green water export and crop export was smaller than those of blue water export. One of the reasons why a large dispersion was shown in blue water export might be that the volume of blue water is much smaller than that of green water. Thus, a small amount of blue water might derive a large change in this plot. However, the main issue in Fig. 1 was that the blue water footprint differed more depending on the exporting country, rather than on the green water footprint. Therefore, the variability of blue water export was larger than that of green water export, and crop export could bring differing impacts on irrigation water by country. |

| Minor Comments 4 | |
|---|---|
| Reviewer's comments | Fig 2: It seems that a subset of countries and links is plotted.
Please clarify how you select those countries and links. |
| Response | We added this explanation about the threshold value in order to display the main network. However, we considered entire countries and all links when we calculated volume, connectivity, and centrality indices of GVWT. Fig. 2 indicated the display of the main connection. |
| page 6, line 26 - page 7, line 2 | The GVWT network includes both the volume of virtual water and the connection among countries. Fig. 2 shows only the main GVWT network of food and feed crops in 2010 using the threshold value of virtual water trade, as we could not display these networks with all links, because it is impossible to distinguish each connection between countries. Therefore, we showed the main links that were over a threshold value of 1.0 Gm³ of total virtual water trade in 2010. Some countries were eliminated from the figure, because they only had connections of virtual water trade that were less than the threshold value. GVWT for food crops has a dispersed network, but GVWT for feed crops is more centralized with a few main exporters, such as the U.S.A., Argentina, Brazil, and China. In other words, the food and feed crop trades have a different structure, and we need to consider not only volume, but also the connectivity of the virtual water trade. |

| Minor Comments 5 | |
|---|---|
| Reviewer's comments | P6 L33: Please clarify how the volume and connectivity are classified into three groups. In particular, what is the threshold? Why the threshold is as it is? |
| Response | We added this explanation on manuscript. |

| | | |
|---|---|---|
| | page 8, line 15 - page 9, line 2 | We considered the vulnerability of virtual water trade to be more related to importers with the larger volume of virtual water import. Therefore, the top 10 percentile of total virtual water import was used as the threshold. After that, we classified these countries into 3 groups, according to the top 1, 5, and 10 percentiles of total virtual water import. The small volume group (I) includes countries that imported above the top 10 percentile and below the top 5 percentile of total virtual water import; the medium group (II) includes the countries that imported over the top 5 percentile and below the top 1 percentile of total virtual water import. Finally, the large volume group (III) includes countries that imported over the top 1 percentile of total virtual water import. In addition, the vulnerable virtual water trade could be related to the connectivity; therefore, we classified the importers into high, medium, and low connectivity groups, using the degree centrality of links. The importers who have a low degree centrality of links could be regarded as relatively vulnerable importers, and we use the maximum degree centrality of links as the standard for evaluating the connectivity of each country. Therefore, the importers who have the upper one third of maximum degree centrality are classified as the high connectivity group (C), and the importers who have the lower one third of maximum degree centrality are classified as the low connectivity group (A). The importers who are classified in the medium connectivity group have a degree centrality between the upper one third and lower one third of maximum degree centrality. When importers are classified into the A-III sector, we considered that they had intensive virtual water import with vulnerable structure. In food crops, the upper 10% of virtual water import, 25.1 Gm³ was set as the threshold value, and the upper 5 and 1 percentiles of virtual water import were 37.3 and 72.9 Gm³, respectively. Therefore, the importers in the large volume group imported over 72.9 Gm³ of virtual water during 2006-2010 through food crops. The maximum value of degree centrality was 0.22. Therefore, the high connectivity group included those importers who had a degree centrality between 0.22 and 0.15. The low connectivity group included those importers who had lower than 0.07 degree centrality. |
| | page 9, line 9 - page 9, line 13 | In feed crops, the upper 10% of virtual water import, 23.8 Gm³ was set as the threshold value, and the upper 5 and 1 percentiles of virtual water import were 42.3 and 103.6 Gm³, respectively. Therefore, the importers in the large volume group imported over 103.6 Gm³ of virtual water during 2006-2010 through food crops. The maximum value of degree centrality was 0.17. Therefore, the high connectivity group included importers who had a degree centrality between 0.17 and 0.11. The low connectivity group included the importers who had lower than 0.06 degree centrality. |

| Minor Comments 6 | | |
|---|---|---|
| Reviewer's comments | | P7 L15-22: Discussion about water savings is irrelevant to this part, which focuses on vulnerability. Please take it out. |
| Response | | We added the new chapter which is "3.2.3 GVWT impacts on water savings in importing countries", and moved the discussion about water savings to this new chapter. |
| | page 9, line 20 - page 10, line 8 | **3.2.3 GVWT impacts on water savings in importing countries**
Virtual water trade could help the importers save water resources by crops import. For example, if the importing country replaces crop import with domestic production, this will be accompanied by additional water use. Table 4 shows the water savings by virtual water import in main importers from 2006 to 2010. China and Japan, respectively saved 24.7 and 18.7 Gm³/yr of green water by crops import, while Egypt and Iran, respectively saved 15.3 and 10.1 Gm³/yr of blue water by crops import, depending on irrigation water for domestic crop production. In particular, Egypt and Iran have few water resources, therefore, the virtual water impacts on water resource savings in these countries might be larger than on other importers.
Accordingly, VWT is a very important issue for these importers; thus the vulnerable structure of VWT could cause water shortage problems to importing countries. For example, in 2010, Russia banned wheat export because of severe drought, and the global wheat price rose. Oxfam Research Reports analysed the impacts of the Russian ban of wheat export on global and local areas in terms of economic impacts (Welton, 2011). Wheat import in Egypt has high dependency on the Russian federation's export, which we regarded as a vulnerable structure, and the insufficient import of crops due to the export ban in the Russian Federation could bring not only economic impacts but also serious water consumption for increasing domestic food production. Chapagatin et al. (2006) found the import of wheat in Egypt contributed to a national water saving of 3.6 Gm³/yr during 1997-2001, which according to the 1959 agreement was about seven percent of the total volume of water to which Egypt was entitled. Fader et al. (2011) also found that some water-scarce countries, such as China and Mexico, the Netherlands and Japan, would need relatively high amounts of water to produce the goods they otherwise import: meaning that they save high amounts of water by importing goods. Therefore, if they stopped importing and exporting agricultural products, these countries would need to use more water in their agricultural sectors (Fader et al., 2011). In other words, a vulnerable trade structure with low connectivity could be one of the main reasons for water shortage problems. |

---

## Referee Report (RR1)

This paper presents an interesting analysis of the global virtual water trade of food and feed crops. They use network analysis method to discuss the connectivity and vulnerability in food trade and feed trade. The methods are valid and the discussions are appropriate.  In this case, I suggest it to be published as is.